# Privacy preserving data sharing method for social media platforms

**Snehlata Yadav**[1,2]*, **Namita Tiwari**[1]

**1** Department of Computer Science and Engineering, Maulana Azad National Institute of Technology, Bhopal, MP, India, **2** Department of Computer Science and Engineering, Government Women's Polytechnic College, Bhopal, MP, India

* yadavsnehlata@gmail.com

## Abstract

Digital security as a service is a crucial aspect as it deals with user privacy provision and secure content delivery to legitimate users. Most social media platforms utilize end-to-end encryption as a significant security feature. However, multimedia data transmission in group communication is not encrypted. One of the most important objectives for a service provider is to send the desired multimedia data/service to only legitimate subscriber. Broadcast encryption is the most appropriate cryptographic primitive solution for this problem. Therefore, this study devised a construction called *anonymous revocable identity-based broadcast encryption* that preserves the privacy of messages broadcasted and the identity of legitimate users, where even revoked users cannot extract information about the user's identity and sent data. The update key is broadcast periodically to non-revoked users, who can obtain the message using the update and decryption keys. A third-party can also revoke the users. It is proven that the proposed construction is semantically secure against IND-ID-CPA attacks and efficient in terms of computational cost and communication bandwidth.

**Data Availability Statement:** All relevant data are within the paper.

**Funding:** The author(s) received no specific funding for this work.

## 1 Introduction

Nowadays, the use of social media [1] has become an essential part of today's life. Previously, short messaging service (SMS) was provided by Global System for Mo-bile Communication (GSM) and Code Division Multiple Access (CDMA) operators. Today, several instant messaging (IM) services offer multimedia data transmission to multiple users simultaneously and facilitate communication with many participants via group chats, which offers a significant advantage over SMS. IM chats [2] also allow the sharing of text messages and attachments, such as image or videos, for direct and group communication. Groups in social media comprise a set of members and contain meta information, such as group title. Based on the IM application and its underlying protocols, groups can be modified either by all the participating users or administrated by some selected users. These applications generate large amounts of data, which are sensitive in nature. If these data are revealed, then there may be many consequences related to the privacy of all associated users. For these applications, effective and

**Competing interests:** The authors have declared that no competing interests exist.

efficient methods must be deployed to ensure data security and confidentiality of user credentials. Consider a scenario where the data owner wants to transmit multimedia data to only a subset of members from the group of users. For this, many cloud providers support a broadcast feature, which is not end-to-end encrypted. Government, healthcare services, social media networks, associations, businesses, and individuals have been gathering personal information for analysis, decision-making and other reasons for decades. For example, health records are used to monitor disease transmission to uncover the secret connection between diseases and their prevention and control. Organizations may, for instance, provide information of their clients to external third parties, such as advertising agencies. Sharing and distributing information in a targeted group while ensuring data privacy and user's identity privacy is critical but has not been given much attention. In addtion to social media platforms, smart cities, the Internet of Things(IoT), cloud environments, Pay TV, and, healthcare systems have the potential for the application for proposed scheme. Consider a scenario of identity-based broadcast encryption in smart city that uses information and communication technologies (ICT) and IoT technologies to improve the life quality of all residents. For example, to identify traffic congestion data of users can be collected by some sensors, and electricity consumption can be monitored by smart meters enabling residents to receive better services than before. Nevertheless, most new devices, such as smart meters, sensors, traffic and surveillance cameras, traffic lights, and, cellphones are wireless, which are very easy to be attacked if communications are not properly encrypted. Another example of healthcare scenario using an anonymous identity-based broadcast encryption, is the health record sharing system for hospitals, which is supplied by a cloud service. Without losing generality, it can be assumed that the system consists of a cloud server, a data owner, and a group of doctors labelled "S". The data owner first encrypts a data for a selected group and then stores the encrypted file in the cloud for sharing. When a doctor leaves the hospital, the server must revoke him from accessing all files. Revoked set is denoted as "R". If the revoked doctors are not in group S, they cannot decrypt the ciphertext after the server conducts revocation. Most importantly, it requires the cloud server to be able to revoke users from a ciphertext without knowing the encrypted file and identities of receivers. For this, the server must have the ability to decrypt the ciphertext. When some identities should be revoked, the server first decrypts the ciphertext and removes them from the original authorized user set. It then re-encrypts the data using the new authorized user set. However, in this trivial solution, the cloud server is able to learn the content and the identity of authorized users who can access the file. A cloud server must revoke the user without knowing the identity of user at a particular time period using time key. Simultaneously, the cloud server must not learn any information pertaining to data owner's health record. Thus, this paper proposes an anonymous revocable identity-based broadcast cryptosystem (`ARIBBE`) that overcomes these challenges through the following:

- The proposed scheme preserves the privacy of legitimate subscribers.

- The primary objective of the proposal method is the retrieval of the function without learning the revocation identity set with ciphertext evolution and protection of data privacy despite revoked user collisions.

- This paper presents a new cryptosystem with ciphertext evolution over cloud, which is appropriate for fine-grained data access control, protects legitimate receiver's identity, and allows a third party to revoke select receivers.

- The construction is semantically secure under BDHP computational assumption and random oracle model.

- The computation cost in the revocation phase is linear to the number of revoked users and revocation is performed using time key.

The remainder of the paper is organized as follows. The Literature review section 2 presents an overview of similar existing schemes. The Preliminaries section 3 provides the definition and the overview of basic ID-based broadcast cryptosystem. The Proposed Scheme section 4 describes the proposed method. The Security Model and Proof Overview section 5 provides the security analysis of the proposed scheme. Finally, the Conclusion section 6 presents the conclusions of the study.

## 2 Literature review

The concept of broadcast encryption (BE) was first proposed by Fiat and Naor [3], however, Boneh *et al.* [4] constructed a broadcast encryption scheme with smaller decryption, encryption key size, shorter encrypted message size, and better computational cost. BE is a cryptographic primitive which provides a solution to the problem of communicating encrypted messages to only legitimate set of users, "S", over an insecure channel. Only legitimate users from S can decrypt the encrypted message. In contrast, revoked users of S would not learn anything about the message. Users who obtain access to the ciphertext are called legitimate subscribers (member of the set S) and unsubscribed users (non-member of the set S) are called revoked users. The broadcast encryption algorithm works on the demarcation of revoked and legitimate users and this partition can vary for each broadcasted message. Broadcast encryption is deployed in two settings: symmetric key and asymmetric key settings. In a symmetric key framework, a key generation center distributes the secret decryption keys to all legitimate users before message transmission phase. For each user, a separate secret key is maintained. In such a scenario, only the broadcaster or the sender acts as the source of a message. The sender shares a common session key with all the subscribed users.

To broadcast an encrypted message to all users in S, message is encrypted using the session key and then, to decrypt it, the legitimate users need the session key, their own secret key, and S to identify all the receivers. In the asymmetric key setting, the broadcast encryption uses public key framework. All users of the set S has a pair of keys, one for encryption function and another for decryption function. The broadcaster and other possible other entities can act as a source of a message, while, only legitimate subscribers or receivers can decrypt and learn the actual message. It also resolves the problem of refreshing the secret keys after any update in the set of legitimate subscribers for symmetric key setting. BE in the public key setting is well studied and can be classified as Fig 1: Identity-based broadcast encryption, attribute-based broadcast encryption, anonymous broadcast encryption, hierarchical broadcast encryption, dynamic broadcast encryption, and distributed broadcast encryption. It has several applications such as in secure email system, digital rights management system, pay TV, database security system, online social network system etc. Identity-based broadcast encryption (IBBE) was

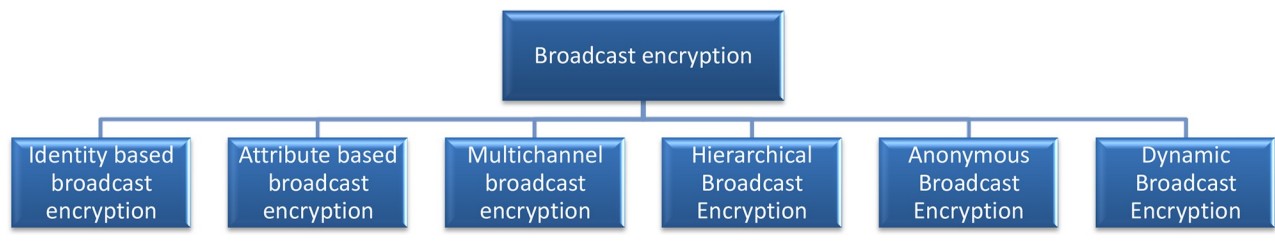

**Fig 1. Broadcast encryption in public key framework.**

first introduced by C. Delerablée [5], which is an extension of identity-based encryption scheme in public key setting. Instead of public keys each legitimate subscriber is identified by their Identity, such as email-id, passport number, or driving license number (arbitrary strings, alphanumeric values, and numerals) etc., are used as encryption keys. It is a practical cryptographic primitive that allows an exponential number of recipients to exchange messages securely. This implies that the public parameter is not correlated by any means to the decryption key of recipients.

Due to high prevalence of IoT technology [6, 7] applications and blockchain technology (BCT), security issues such as identity authenticity and data privacy are becoming increasingly important concern. Fan et al. [8] integrated IoT with BCT and proposed efficient authentication and secure data sharing scheme. This scheme achieves a proper tradeoff between security and performance, compared with other schemes of IoT but the only drawback is, it doesn't provide an anonymous authentication. Blockchain [9] along with attribute based searchable encryption offers decentralized and computationally efficient construction. For e-health [10] proposed secure and energy-efficient IoT model. It enables secure transmission and retrieval of biomedical images over IoT networks. All aforementioned schemes [6, 8–10] do not provide an anonymous authentication property.

Various online social networks [11–13] (e.g. WhatsApp, Twitter, Instagram, and Facebook) make the distribution of user's real time data between multiple users over the same and different networks very easy. The ease of use, faster transformation, and cost-effectiveness make online social networks an efficient method of communication and information sharing [14, 15]. Many researchers have extensively analyzed the impact of social media [16] on information sharing. However, these schemes does not ensure the anonymity of the receivers. To address this problem [4] has come up with the anonymous broadcast encryption in public key setting and the issue of anonymity has been studied extensively in schemes, [4, 17–24]. Many applications e.g., vehicular ad-hoc network(VANET) [18] use computationally efficient privacy preserving anonymous authentication scheme based on the use of anonymous certificates and signatures which is an important component of IoT. This scheme is efficient in terms of certificate and signature verification cost and providing anonymity. Azees *et al.* [25, 26] proposed an efficient conditional privacy preserving scheme for VANET using bi-linear pairing. The scheme provides better efficiency in terms of fast verification on certificates, signatures, preserves anonymity among vehicle entities and revokes the privacy of misbehaving vehicles and provides conditional privacy in a computationally efficient manner. VANET entities become anonymous to each other until they are revoked from the VANET system. In the e-healthcare domain, data privacy and security of electronic health records are the most prominent challenges with cryptographic primitives playing a vital role in providing privacy and secure access. Chen et al. [27] presented a comprehensive review of privacy preserving methods in this domain. However, none of these schemes are able to achieve anonymity and revocation simultaneously with respect to the time key. This paper attempt to solve this problem.

## 2.1 System model

Consider an online e-healthcare system, such as that presented in Fig 2, where the data related to patients is collected and uploaded to the centralized storage server, for example, a cloud server. The patient's data must be secure and the privacy (identity) of the patient must be preserved. If this is not taken seriously, the patients may suffer the consequences of having their medical records leaked online. Recently millions of user's data has been compromised.

In online e-healthcare system, a patient acts as data owner and may they choose to share their medical data with various medical professionals or related personnel, such as

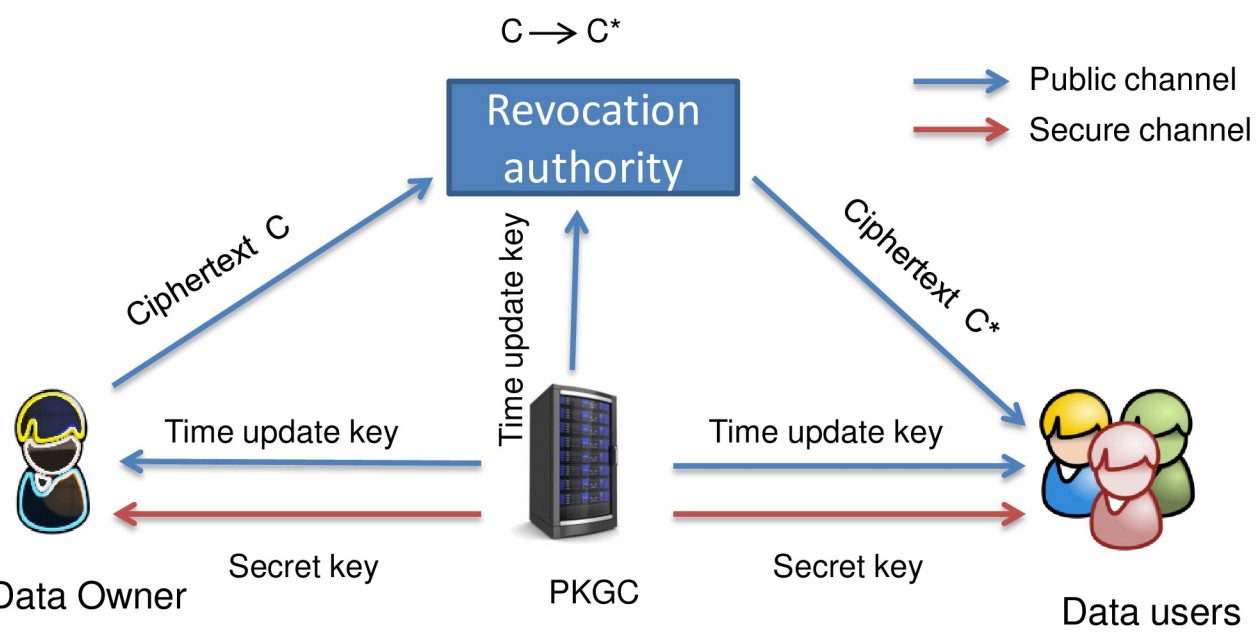

**Fig 2. A system model for online data sharing.**

1. to various doctors for taking opinion for his medical case.

2. government offices for providing information as they are working in that office many a times, it is necessary to share medical records of the employee with employer.

3. and insurance agencies for medical-claim disbursement purposes.

In other words, if a data owner chooses to share their medical records to various receivers where no receiver is able to learn other receivers' identity. If some receivers are revoked then they cannot learn any message by combining their keys. This revocation list is based on time update keys. Most of the time, online healthcare data resides in shared environments, thus, ensuring that the data is shared and accessed in a secure manner on the cloud and access is a non-trivial task. One way to share data among group of legitimate subscribers is identity-based broadcast encryption. The privacy of the recipients is an important issue to be addressed in broadcast encryption schemes. There are scenarios where revealing the identity of receivers may result in a threat to the subscribed users. To some extent, this issue has been taken into consideration by several past researchers [4, 28–31]. However, whilst these schemes did show data sharing capabilities including receiver anonymity, they did not provide a solution to the problem of when some receivers are revoked from the original subscriber set. Encrypting the message again for the newly formed subscriber set after the revoked user is a trivial but impractical solution. The notion of *Recipient revocable identity-based broadcast encryption* (R-IBBE) provides efficient solution to this problem.

### 2.1.1 Design goals.

- No probabilistic polynomial-time (PPT) adversary is able to recover plaintext from ciphertext after revocation. This scheme is aimed at being secure against chosen plaintext attacks.

- The scheme is collusion resistant. More particularly, if the maximum receivers is set as one, the resultant scheme is an anonymous revocable IBE scheme with timestamp revocation.

- The receiver set is anonymous i.e., the identities of the receivers are hidden from outside world.

- It should be difficult for the cloud server to retrieve the identities of receivers from $C^*$.

- The computational cost of decryption is independent of total number of receivers.

- The public parameters in the proposed scheme is linear in the maximum size of the privileged identity set.

## 2.2 Design issues of the `IBBE` schemes

There are various essential design issues for the construction of `IBBE` cryptographic primitives. These are briefly discussed below.

**2.2.1 Security model.** In the basic security model of `IBBE` schemes, an adversary is allowed to obtain the secret keys $S_k$ for a specific set, S, of identities of $k$-subscribers. The adversary can-not break the encryption scheme for some another identity set S′ or for some other subscriber's identity. The security model also allows adversary to obtain secret keys corresponding to an identity ID $\subseteq$ S. Based on the targeted identity set by an adversary, two types of security notions emerge, namely: selective security and adaptive security. Selective security is a *weaker* security notion that allows an adversary to specify a target recipient set before learning about the public parameters, however, here the adversary is restricted in raising encryption queries.

Whereas, in the adaptive security, a *stronger* security notion, allows an adversary to specify target recipient set adaptively. The adversary is able to corrupt identities even before it knows about the recipient identity set. Semantic security and chosen ciphertext security notions are defined as stronger security notions and the schemes based on these security notions are highly desirable.

**2.2.2 Computational complexity assumption.** Security of cryptographic primitives are based on computationally hard problems. There are several standard computation problems based on which most of the public-key cryptosystems are constructed. Apart from standard computational complexity assumption, many cryptosystems are constructed considering reduction of a problem to another problem. Bilinear pairings or mappings are employed in most of the public key broadcast encryption schemes. Moreover, lattice-based or code-based schemes can also serve as a candidate that can resist attacks using quantum computers.

**2.2.3 Header size.** The ciphertext consists of a header (*hdr*) and a session key *K*. The original message to be transmitted is encrypted with the help of this session key. Moreover, the header and ciphertext contains some additional knowledge that facilitates only privileged recipients to retrieve the session key and then subscribed users obtain the message with the help of this recovered session key. It is desirable to have a compact size header to reduce communication overhead.

**2.2.4 Key size.** For lightweight cryptographic applications, such as in IoT, Industry-IoT, and wireless sensor networks, a scheme is required where the size of keys and computational cost are kept to a minimum because of storage constraints. Therefore, it is preferred to use shorter key and header size.

**2.2.5 Pairing choice.** There exist many pairing based efficient construction of `IBBE` schemes. Pairing based construction requires a function defined as $\hat{e} : \tilde{\mathbb{G}}_1^+ \times \tilde{\mathbb{G}}_2^+ \to \tilde{\mathbb{G}}_T^*$, where $\tilde{\mathbb{G}}_1^+$ and $\tilde{\mathbb{G}}_2^+$ are additive groups and $\tilde{\mathbb{G}}_T^+$ is a multiplicative group for some large prime $q$. Pairing based constructions can be categorized into three types: Type-I, Type-II, and Type-III, as described below [32, 33].

1. If both the input cyclic groups are same (i.e. $\tilde{\mathbb{G}}_1^+ = \tilde{\mathbb{G}}_2^+$) then the pairing is considered *Symmetric pairing or Type-I pairing*.

2. *If both the input cyclic groups are not same ($\tilde{\mathbb{G}}_1^+ \neq \tilde{\mathbb{G}}_2^+$) and there exist precisely a computable homomorphism $\phi : \tilde{\mathbb{G}}_2^+ \to \tilde{\mathbb{G}}_1^+$ then the pairing is considered to be Type-II pairing.*

3. *However, if both the input cyclic groups are not same and there does not exist a computable homomorphism $\phi : \tilde{\mathbb{G}}_1^+ \to \tilde{\mathbb{G}}_2^+$ then the pairing is called Asymmetric pairing or* Type-III *pairing.*

All types of pairing have been well studied in literature [34, 35] with the consensus being that *Type-III* is more suitable to use because it provides designs based on this are most efficient, secure, and have more compact parameter sizes. In addition to bi-linear pairing, lattice is also one of the most powerful tools for constructing post quantum cryptographic primitives.

**2.2.6 Additional properties.** Several efficient algorithms of IBBE have been constructed based on various complexity assumptions, security models, compact size of header, and keys. Apart from all these parameters, some additional properties are also required.

1. **Anonymity property**: This states that the adversary cannot obtain set of receivers from the ciphertext. Leakage of the recipient set reveals the subscriber's information which is a major security concern and may cause a personal attack to receivers such as trolling, bullying, etc. Previous schemes have focused on the confidentiality of the message while the recipient set was openly known to adversaries. Preserving privacy is a serious concern in designing cryptographic algorithms, and therefore, various broadcast encryption schemes consider this issue such as Anonymous-IBBE [21], Private broadcast encryption [4], Attribute-based broadcast encryption (hidden policy) [22, 23], outsider anonymous BE [36], lattice-based BE [24].

2. **Revocation**: This is the intrinsic property of basic broadcast encryption techniques. Consider a scenario where the broadcast encryption setup algorithm and key generation algorithm generate all the parameters as per the construction. The system has reported some malicious receiver or somehow the legitimate receiver's decryption key is leaked. Therefore, in these circumstances, a receiver needs to be revoked. Schemes with revocation property are preferred in the construction of systems.

## 3 Preliminaries

This section first describes a relevant IBBE cryptosystem followed by a Revocable identity-based broadcast encryption scheme. Secondly, the basic concepts on bilinear pairing and decisional BDHE assumption are also presented. The notations and acronym used in the paper are described in Table 1.

### 3.1 Identity-based broadcast encryption scheme

The notion of IBBE construction as presented in [5] is described here. An identity-based broadcast encryption method consists of ensemble of three probabilistic algorithms (IBBE. setup, IBBE.extract, IBBE.Enc) and one deterministic algorithm (IBBE.Dec).

1. IBBE.setup $(1^\lambda, m) \to$ (PP, msk). $\lambda$ is considered a security parameter in the algorithm, and the maximum count of identities in a privileged recipient set, $m$, is taken as input to produce the public parameter PP and a master secret key msk. Further, $\mathcal{I}$ denotes the identity space, and $\mathcal{K}$ represents the key space.

**Table 1. Notations.**

| Notation | Description |
|---|---|
| IBBE | Identity based broadcast encryption |
| $\lambda$ | Security parameter |
| $\tilde{P}P$ | Public Parameter |
| $\mathcal{I}$ | Identity space |
| S | Recipient set |
| ID | Identity of receiver |
| $\mathbb{G}_1^+, \mathbb{G}_2^+, \mathbb{G}_T^*$ | Cyclic groups |
| M | Plaintext |
| RL | Revocation List |
| $\mathcal{C}, \mathcal{A}$ | Algorithms |
| $\tilde{P}_{pub}$ | Public key |
| msk | Master secret key |
| $D_{SKID}$ | Decryption key for identity |

2. IBBE.extract (ID, msk) → $D_{SKID}$. Taking an msk and an identity vector ID as inputs, IBBE.extract produce a decryption key $D_{SKID}$ corresponding to the identity ID.

3. IBBE.Enc (PP, M, S ⊆ $\mathcal{I}$) → (C, K). This randomized algorithm produces a pair (C, K), where C is the ciphertext and K ∈ $\mathcal{K}$ is a session key, on the input of public parameters PP, legitimate recipient set of size |S| ≤ *m*, and a message to be sent M ∈ $\mathcal{M}$ from the message space $\mathcal{M}$.

4. IBBE.Dec (ID, PP, S, $D_{SKID}$, C) → M. This deterministic algorithm recovers the message M on taking public parameter PP, legitimate recipients set S = {$ID_1$, . . ., $ID_l$}, identity ID, secret key $D_{SKID}$, and ciphertext C as input. If the message is not recovered then a bottom symbol ⊥ is produced.

One of the essential requirements of IBBE scheme is that for every identity ID ∈ S output by IBBE.extract algorithm is a decryption key $D_{SKID}$ IBBE. enc(PP, M, S ⊆ $\mathscr{I}$) → (C, K) then IBBE.Dec(ID, PP, S, $D_{SKID}$, C) → M with certain probability. The ID-based encryption scheme is a particular instance of IBBE if size of recipient set is 1, i.e. |S| = 1.

## 3.2 Revocable identity-based broadcast scheme

An extension of identity-based broadcast cryptosystem that allows subscribed receivers to be revoked is called revocable identity-based broadcast encryption R-IBBE [37]. This extension facilitates a legitimate subscriber with an ID to be revoked if their credentials are expired or leaked. In R-IBBE, each user of a recipient set obtains a decryption key from the private key generation center/authority (PKGC) that is related with the user's ID. Once the system is configured, the key generation authority periodically updates revoked recipient set RL with respect to time T and then broadcasts the update key for the remaining non-revoked recipients. The generation of the update key depends on RL and T.

If a legitimate user does not revoke at time T when the update key has been issued then the user can generate their own decryption key corresponding to ID. With the use of decryption key corresponding to their ID and time T, the legitimate recipient is able to decrypt a ciphertext for receiver $ID_p$ and time $T_c$ only if ID = $ID_p$ and T = $T_c$ holds.

- If ID ∈ RL then it outputs a pair of decryption key for identity ID and time T.

- If ID ∉ RL then it outputs ⊥ with all but negligible probability.

- If $(ID_p = ID) \wedge (T_p = T)$, then Decryption algorithms outputs M

- If $(ID_p \neq ID) \vee (T_p \neq T)$, then Decryption algorithms outputs ⊥ with all but negligible probability.

### 3.3 Bilinear pairing based on prime order groups

Assume that $\tilde{\mathbb{G}}_1^+$ and $\tilde{\mathbb{G}}_2^+$ are cyclic groups of prime order $q$ with the binary operation being addition, and $\tilde{\mathbb{G}}_T^*$ denotes a multiplicative group of same prime order $q$. Bilinear pairing is a mapping or simply a function $\hat{e} : \tilde{\mathbb{G}}_1^+ \times \tilde{\mathbb{G}}_2^+ \to \tilde{\mathbb{G}}_T^*$ that satisfy the following properties as mentioned below.

1. **Bilinearity**—$\hat{e}(aP, bQ) = \hat{e}(P, Q)^{ab} = \hat{e}(bP, aQ)$, where $\forall P \in \tilde{\mathbb{G}}_1^+, \forall Q \in \tilde{\mathbb{G}}_2^+$ and $\forall a, b \in \mathbb{Z}/q\mathbb{Z}$.

2. **Non-degeneracy**—$\hat{e}(P, Q)$ is a generator element of $\tilde{\mathbb{G}}_T^*$ i.e. $\tilde{\mathbb{G}}_T^* = \langle \hat{e}(P, Q) \rangle$, where $\hat{e}(P, Q) = 1_{\tilde{\mathbb{G}}_T^*}, \forall Q \in \tilde{\mathbb{G}}_2^+$ iff $P = 1_{\tilde{\mathbb{G}}_1^+}$ and similarly $\hat{e}(P, Q) = 1_{\tilde{\mathbb{G}}_T^*} \forall P \in \tilde{\mathbb{G}}_1^+$ if and only if $Q = 1_{\tilde{\mathbb{G}}_2^+}$.

3. **Computability**—A pairing is defined as computable if there exist a polynomial runtime algorithm that can evaluate the expression $\hat{e}(P, Q), \forall P \in \tilde{\mathbb{G}}_1^+, \forall Q \in \tilde{\mathbb{G}}_2^+$ and $\forall a, b \in \mathbb{Z}/q\mathbb{Z}$, correctly.

### 3.4 Hardness assumptions

Let $\mathcal{G} = (\tilde{\mathbb{G}}_1^+, \tilde{\mathbb{G}}_T^*, \hat{e}, q)$ be a Type-I bilinear mapping with a generator $P \in \tilde{\mathbb{G}}_1^+$. Bilinear Diffie–Hellman (BDH) hardness assumption states that given a tuple $(P, aP, bP, cP)$ for some unknown $a, b, c \in \mathbb{Z}/q\mathbb{Z}$ as input then find the output $D = \hat{e}(P, P)^{abc} \in \tilde{\mathbb{G}}_T^*$.

**Assumption 1**. Let *Adversarial* algorithm $\mathcal{A}$ be a $\tau$−time algorithm that receives an input challenge for *BDH* problem and produces a decision bit $\beta \in \{0, 1\}$ as output. $\mathcal{A}$ has advantage $\epsilon$ in solving bilinear Diffie-Hellman problem when

Adv $= |\Pr[\mathcal{A}(P, aP, bP, cP) = \hat{e}(P, P)^{abc}]| \geq \epsilon$, where the probability is over the random choices of $D \in \tilde{\mathbb{G}}_T^*$, random bits consumed by $\mathcal{A}$, random choice of $\alpha \in \mathbb{Z}/q\mathbb{Z}$ and the random choices of generator $P$ of $\tilde{\mathbb{G}}_1^+$, respectively.

**Definition 1**. *The assumption 1 is tenable in* $\tilde{\mathbb{G}}_1^+$ *if no τ-time algorithm has an advantage of at least $\epsilon$ in solving the* BDH *assumption over* $\tilde{\mathbb{G}}_1^+$.

## 4 Proposed scheme

The proposed construction considers the idea of revocation of subscribed receivers used in the `IBE` scheme of [38] and the ID-based broadcast encryption scheme [37].

**Definition 2**. *The anonymous revocable identity-based broadcast encryption (`ARRIBE`) scheme is associated with message space $\mathcal{M}$, identity space $\mathcal{I}$, and time space $\mathcal{T}$. The protocol is an ensemble of probabilistic algorithms, namely,* `ARIBBE.setup, ARIBBE.Genkey,`

*ARIBBE.Timekey, ARIBBE.encrypt, ARIBBE.decrypt, and ARRIBE.
revoke that are described as follows.*

1. `ARIBBE.setup` $(1^\lambda, m)$. This algorithm considers security parameter $\lambda$ and $m$ as the maximum count of identities of users as input parameters. It generates public parameter PP, an empty revocation list RL, and msk. This algorithm is executed by the PKGC.

2. `ARIBBE.Genkey` (PP, msk, ID). This key generation algorithm takes $ID \in \mathcal{I}$, the master secret key msk, and the public parameter PP as inputs. It generates a decryption key $D_{SKID}$ corresponding to the ID as output. Trusted authority runs this algorithm.

3. `ARIBBE.Timekey` (PP, msk, ID, T). This algorithm generates a time update key, and the PKGC broadcasts it periodically.

4. `ARIBBE.encrypt` (PP, M, ID). This algorithm takes a message M to be broadcasted, public parameter PP, and a set of identities S = $\{ID_1, \ldots, ID_l\}$ as input. It produces a ciphertext C as output and the sender of data executes this algorithm.

5. `ARRIBE.revoke` (PP, RL, C). The service provider executes this algorithm. Upon taking the public parameter PP, a ciphertext C, and a revocation identity set RL = $\{ID_1, \ldots, ID_t\}$ as inputs, a new revocation list is produced.

6. `ARIBBE.decrypt` (PP, C, $D_{SKID}$). Upon taking the public parameter PP, a ciphertext $\mathcal{C}$, and the corresponding decryption key $D_{SKID}$ as inputs, it generates the original message M only if ID $\in$ S and ID $\neq$ RL. The receiver of the data runs this algorithm.

## 4.1 Proposed construction

The proposed scheme is ensemble of one deterministic algorithm `ARIBBE.decrypt`, and four randomized algorithm (`ARIBBE.setup, ARIBBE.Genkey, ARIBBE.Time-key, ARIBBE.encrypt`).

1. `ARIBBE.setup` $(1^\lambda, m)$

(a). Randomly select $\tilde{\mathcal{G}} = (\tilde{\mathbb{G}}_1^+, \tilde{\mathbb{G}}_T^*, \hat{e}, q)$.

(b). Given $\hat{e} : \tilde{\mathbb{G}}_1^+ \times \tilde{\mathbb{G}}_1^+ \rightarrow \tilde{\mathbb{G}}_T^*$, where $\tilde{\mathbb{G}}_1^+ = \langle \tilde{P} \rangle$.

(c). Set message space $\mathcal{M} = \{0, 1\}^n$.

(d). Select a k-bit prime number $q > 3$.

(e). Randomly select an element s from $\mathbb{Z}/q\mathbb{Z}$.

(f). Now, set $\tilde{P}_{pub} = s\tilde{P}$.

(g). Define the following hash functions:

$$
\begin{aligned}
\tilde{H} \quad &: \{0, 1\}^n \rightarrow \tilde{\mathbb{G}}_1^+ \\
\tilde{H}_1 \quad &: \tilde{\mathbb{G}}_T^* \times \{0, 1\}^n \rightarrow \mathbb{Z}_p \\
\tilde{H}_2 \quad &: \tilde{\mathbb{G}}_T \times \{0, 1\}^n \rightarrow \tilde{\mathbb{G}}_1^+ \\
\tilde{H}_3 \quad &: \tilde{\mathbb{G}}_T^* \times \{0, 1\}^n \rightarrow \tilde{\mathbb{G}}_1^+ \\
\tilde{H}_4 \quad &: \tilde{\mathbb{G}}_T^* \times \{0, 1\}^n \rightarrow \tilde{\mathbb{G}}_1^+
\end{aligned}
$$

(h). Public parameters are:

$$PP = (\tilde{\mathcal{G}}, \tilde{P}, \tilde{P}_{pub}, \tilde{H}, \tilde{H}_1, \tilde{H}_2, \tilde{H}_3, \tilde{H}_4) \tag{1}$$

2. `ARIBBE.Genkey` $(PP, msk, \tilde{ID})$

(a). Evaluate $ID = s \cdot \tilde{H}(\tilde{ID})$.

3. `ARIBBE.Timekey` $(PP, msk, ID, T)$

(a). The private key generation center, to generate time keys, evaluate the following:

$$Q_{ID,T} = \tilde{H}_4(ID, T)$$
$$TK_{ID,T} = s \cdot Q_{ID,T}$$
$$\text{Revocation list} = RL \cup \{ID, T\}$$

4. `ARIBBE.encrypt` $(PP, M, ID)$

(a). Given public parameter PP, a plaintext M and a set of identities $S = \{ID_1, \ldots, ID_l\}$, select a dummy recipient $ID_0 \in S$.

(b). Select $(r_1, r_2, r_3 \in_R \mathbb{Z}/q\mathbb{Z})$ and $K_1 \in_R \tilde{\mathbb{G}}$.

(c). Compute,

$$C_1 = K_1 + M \tag{2}$$

$$C_2 = r_1 \cdot \tilde{P} \tag{3}$$

$$C_3 = r_2 \cdot \tilde{P} \tag{4}$$

$$C_4 = r_3 \cdot \tilde{P} \tag{5}$$

(d). For $i = [0, n]$ evaluate,

$$x_i = \tilde{H}_1(\hat{e}(x_A)^{r_1} \oplus \hat{e}(x_B)^{r_1}, ID_i),$$
$$\text{where} \quad x_A = (\tilde{H}(ID_i), \tilde{P}_{pub}) \text{and}$$
$$x_B = (\tilde{H}(ID_{i,T}), \tilde{P}_{pub})$$

and construct the polynomial function as,

$$
\begin{aligned}
f_1(x) &= \prod_{j=0, j \neq i}^{n} \left( \frac{x - x_j}{x_i - x_j} \right) \\
&= \sum_{j=0}^{n} (a_{i,j} x^j \bmod q) \\
\delta_i &= \tilde{H}_2(\hat{e}(x_A)^{r_2}, ID_i) \\
\tilde{\delta}_i &= \kappa_1 + \tilde{H}_3(\hat{e}(x_A)^{r_2}, ID_i)
\end{aligned}
$$

compute,

$$
\begin{aligned}
Q_i &= \sum_{j=0}^{n} a_{j,i} \delta_j \\
U_i &= \sum_{j=0}^{n} a_{j,i} \tilde{\delta}_j \\
C &= (C_1, C_2, C_3, C_4, r_1, [Q_i]_{i=0}^{n}, [U_i]_{i=0}^{n})
\end{aligned}
$$

5. ARIBBE.revoke (PP, RL, C)

   (a). If $RL = \emptyset$ then set $C^* = C$

   (b). Compute $C_1^* = \kappa_2 + C_1$, where $\kappa_2 \in_R \hat{\mathbb{G}}$.

   (c). For all $ID_i \in RL$ compute,

$$
x_i = \tilde{H}_1 \hat{e}(x_A)^{r_1} \oplus \hat{e}(\tilde{H}_4 \hat{e}(x_B)^{r_1}, ID_i)
$$

   and evaluate,

$$
\begin{aligned}
g(x) &= \prod_{i=0}^{t} (x - x_i) \\
&= \sum_{i=0}^{t} b_i \cdot x^i \bmod q
\end{aligned}
$$

   (d). For $i = 1, 2, \cdots, t$, evaluate $Q_i^* = Q_i + b_i \kappa_2$ and set,

$$
\mathcal{C}^* = (C_1^*, C_2, C_3, [b_i]_{i=0}^{i=t-1}, [Q_i]_{i=0}^{i=n}, [U_i]_{i=0}^{n}) \tag{6}
$$

6. ARIBBE.decrypt (PP, C, $D_{SKID}$)

   Given $C^*, \tilde{PP}, ID_i$ and $D_{SKID_i}$, the algorithm recovers original message in following computational steps:

$$
\begin{aligned}
x_i &= \tilde{H}_1(\hat{e}(D_{SKID_i}, C_2) \oplus \hat{e}(TK_{ID,T}, C_2), ID_i) \\
g(x_i) &= \prod_{i=0}^{t} (x - x_i) + x_i^t \bmod q
\end{aligned}
$$

if $g(x_i) = 0$ abort the process, otherwise, compute further,

$$U = U_0 + x_i U_1 + x_i^2 U_2 + \cdots + x_i^n U_n \tag{7}$$

$$Q = Q_0^* + x_i Q_1^* + x_i^2 Q_2^* + \cdots + x_i^n Q_n^* \tag{8}$$

$$+x_i^{t+1} Q_{t+1} + \cdots + x_i^n Q_n \tag{9}$$

then using decryption key $d_{SKID}$, retrieve the session keys $\kappa_1', \kappa_2',$

$$\begin{aligned}
\kappa_1' &= U - \tilde{H}_3(\hat{e}(D_{SKID_i}, C_4), ID_i) \\
\kappa_2' &= g(x_i)^{-1} \cdot (Q - \tilde{H}_2(\hat{e}(C_3, D_{SKID_i}), ID_i))
\end{aligned}$$

and recovers the original message as $M = C_1^* - \kappa_1' - \kappa_2'$. When the identity $ID_i \in S$ and $ID_i \notin RL$ then $\kappa_1' = \kappa_1$ and $\kappa_2' = \kappa_2$ holds.

**4.1.1 Correctness.** For any $ID_i \in S$,

$$\begin{aligned}
\tilde{x}_i &= \tilde{H}_1(\hat{e}(D_{SKID_i}, C_2) \oplus \hat{e}(TK_{ID,T}, C_2), ID_i) \\
&= \tilde{H}_1(\hat{e}(s.\tilde{H}(ID_i), r_1\tilde{P}) \oplus \hat{e}(s.Q_{ID_i,T}, r_1\tilde{P}), ID_i) \\
&= \tilde{H}_1(\hat{e}(\tilde{H}(ID_i)^{r_1}, s\tilde{P}) \oplus \hat{e}(Q_{ID_i,T}, s\tilde{P})^{r_1}, ID_i) \\
&= \tilde{H}_1(\hat{e}(\tilde{H}(ID_i)^{r_1}, \tilde{P}_{pub}) \oplus \hat{e}(\tilde{H}_4(ID_i, T), \tilde{P}_{pub})^{r_1}, ID_i) \\
&= \tilde{H}_1(\hat{e}(\tilde{H}(ID_i)^{r_1}, \tilde{P}_{pub}) \oplus \hat{e}(\tilde{H}_4(ID_i, T), \tilde{P}_{pub})^{r_1}, ID_i) \\
&= x_i
\end{aligned}$$

After obtaining the $x_i$ with the help of decryption key, evaluate,

$$\begin{aligned}
Q &= Q_0^* + x_i Q_1^* + \cdots + x_i^t Q_t^* + x_i^{t+1} Q_{t+1} + \cdots + x_i^n Q_n \\
&= f_0(x_i)\delta_0 + f_1(x_i)\delta_1 + f_2(x_i)\delta_2 + \cdots + f_n(x_i)\delta_n + g(x_i)u \\
&= \delta_i + g(x_i)u \\
U &= U_0 + x_i U_1 + x_i^2 U_2 + \cdots + x_i^n U_n \\
&= f_0(x_i)\tilde{\delta}_0 + f_1(x_i)\tilde{\delta}_1 + f_2(x_i)\tilde{\delta}_2 + \cdots + f_n(x_i)\tilde{\delta}_n \\
&= \tilde{\delta}_i
\end{aligned}$$

Note that $f_i(x_j) = 1$ if $i = j$ and $f_i(x_j) = 0$, otherwise. Now, $\kappa_1$ is evaluated as,

$$
\begin{aligned}
\kappa_1' &= U - \tilde{H}_3(\hat{e}(D_{\mathrm{SKID}_i}, C_4), \mathrm{ID}_i) \\
&= \tilde{\delta}_i - \tilde{H}_3(\hat{e}(s \cdot \tilde{H}(\mathrm{ID}_i), r_3 \cdot \tilde{P}), \mathrm{ID}_i) \\
&= \kappa_1 + \tilde{H}_3(\hat{e}(\tilde{H}(\mathrm{ID}_i), \tilde{P}_{\mathrm{pub}})^{r_3}, \mathrm{ID}_i) \\
&\quad - \tilde{H}_3(\hat{e}(s \cdot \tilde{H}(\mathrm{ID}_i), r_3 \cdot \tilde{P}), \mathrm{ID}_i) \\
&= \kappa_1 + \tilde{H}_3(\hat{e}(\tilde{H}(\mathrm{ID}_i), \tilde{P}_{\mathrm{pub}})^{r_3}, \mathrm{ID}_i) \\
&\quad - \tilde{H}_3(\hat{e}(\tilde{H}(\mathrm{ID}_i), s \cdot \tilde{P})^{r_3}, \mathrm{ID}_i) \\
&= \kappa_1 + \tilde{H}_3(\hat{e}(\tilde{H}(\mathrm{ID}_i), \tilde{P}_{\mathrm{pub}})^{r_3}, \mathrm{ID}_i) \\
&\quad - \tilde{H}_3(\hat{e}(\tilde{H}(\mathrm{ID}_i), \tilde{P}_{\mathrm{pub}})^{r_3}, \mathrm{ID}_i) \\
&= \kappa_1
\end{aligned}
$$

For all $\mathrm{ID}_i \in S$ and $\mathrm{ID}_i \notin RL$, we have $g(x_i) \neq 0$, $\mathcal{H} = g(x_i)^{-1}$ and obtains $\kappa_2$ by computing

$$
\begin{aligned}
\kappa_2' &= \mathcal{H} \cdot (Q - \tilde{H}_2(\hat{e}(D_{\mathrm{SKID}_i}, C_3), \mathrm{ID}_i)) \\
&= \mathcal{H} \cdot (\delta_i + g(x_i)u - \tilde{H}_2(\hat{e}(s \cdot \tilde{H}(\mathrm{ID}_i), r_2\tilde{P}), \mathrm{ID}_i)) \\
&= \mathcal{H} \cdot (\tilde{H}_2(\hat{e}(x_A)^{r_2}, \mathrm{ID}_i) + g(x_i)\kappa_2 - \tilde{H}_2(\hat{e}(x_A), \mathrm{ID}_i)) \\
&= g(x_i)^{-1} \cdot (g(x_i)\kappa_2) \\
&= \kappa_2
\end{aligned}
$$

After retrieving $\kappa_1$ and $\kappa_2$, the message is recovered as,

$$
\begin{aligned}
& C_0{}^* - \kappa_1' - \kappa_2' \\
\text{or,} \quad &= M + \kappa_1 + \kappa_2 - \kappa_1 - \kappa_2 \\
\text{or,} \quad &= M
\end{aligned}
$$

## 5 Security model and proof overview

The formalization of selective-ID security against chosen-plaintext attack the proposed `ARIBBE` scheme is presented here. The data moderator or owner sends the encrypted data to the server. Therefore, apart from the essential requirement that the ciphertext $\mathcal{C}^*$ preserves the message privacy as well as the receiver privacy from insiders and outsiders, the ciphertext must not reveal any information about the message and should also maintain the receiver privacy on the server. Specifically, the security requirements are as follows.

1. No probabilistic polynomial-time (PPT) algorithm can distinguish between the message and identity set contained in ciphertext without having a valid decryption key for a valid identity $\mathrm{ID} \in S$.

2. No PPT algorithm can distinguish a message M contained in ciphertext $C^*$ without a valid decryption key for a valid identity $\mathrm{ID} \in S\backslash R$.

3. No PPT algorithm can distinguish between a revoked identity set RL in $C^*$ without a valid decryption key in RL.

The indistinguishability of the `ARIBBE` cryptosystem is defined as a sequence of messages communicated between an *Adversary* $\mathcal{A}$ and a *Challenger* $\mathcal{C}$ described as a game shown below [39].

- **Setup**. Challenger algorithm $\mathcal{C}$ takes $\lambda$ (security parameter) as an input to generate a master public key $\tilde{P}P$ and a master secret key msk, which is then transmitted to the adversarial algorithm $\mathcal{A}$.

- **Phase I**. $\mathcal{A}$ adaptively issues queries to a decryption key generation oracle for any given identity. Therefore, $\mathcal{C}$ executes the Genkey algorithm to produce decryption key $D_{SKID}$.

- **Challenge**. In this phase, $\mathcal{A}$ transmits two equal length distinct messages $M_0$ and $>M_1$ and a challenge identity set $S^* = (ID_1, ID_2, \ldots, ID_n)$ with the only constraint that $\mathcal{A}$ should not query the decryption key for any $ID_i \in S^*$ in **Phase I**. $\mathcal{C}$ randomly selects a bit $\beta \in \{0, 1\}$ and produces the challenge ciphertext $C^*$ for plaintext $M_\beta$ under identity set $S^*$ and then transmits $C^*$ to $\mathcal{A}$. In **Phase II**, $\mathcal{A}$ issues decryption key queries adaptively similar to Phase I.

- **Guess**. Eventually, $\mathcal{A}$ comes up with its guess $\beta' \in \{0, 1\}$ and if $\beta = \beta'$ then $\mathcal{A}$ wins the game with invariable probability $\epsilon$. $\mathcal{A}'$s advantage in winning the game is defined as:

$$\mathsf{Adv}_{\mathcal{A}}^{\mathsf{IND-CPA}}(\lambda) = \left| \Pr[\beta' = \beta] - \frac{1}{2} \right| \qquad (10)$$

**Definition 3**. *Semantic security of the proposed* `ARIBBE` *scheme achieves* $(\tau, \epsilon)$ *IND-CPA security, if any PPT adversarial algorithm* $\mathcal{A}$ *has invariable negligible advantage* $\mathsf{Adv}$.

In Table 2, the proposed construction is compared with other similar broadcast encryption schemes. Here, Ano is anonymity property, RO is random oracle, $|PK|$ denotes public key parameters, $|CT|$ is the size of ciphertext, S denotes the receiver set, $\hat{n}$ is the length of an identity bit string, TKU is the time key update, Rev is the revocability and $m$ denotes the total number of subscribed users in the system. The scheme [40] does not use pairing for constructing the scheme. We have obtained this for comparing the anonymity and revocation property. Some schemes are anonymous but not revocable, while this scheme ensures both. Further,

**Table 2. Comparison of various `IBBE` schemes.**

| Scheme | Ano | RO | $|PK|$ | $|CT|$ | DT | TKU | Rev |
|---|---|---|---|---|---|---|---|
| [5] | ✗ | ✓ | $O(m)$ | $O(1)$ | $O(S)$ | ✗ | ✗ |
| [41] | ✗ | ✗ | $O(m)$ | $O(1)$ | $O(1)$ | ✗ | ✗ |
| [42] | ✗ | ✗ | $O(\hat{n})$ | $O(S)$ | $O(\hat{n})$ | ✗ | ✗ |
| [43] | ✗ | ✓ | $O(m)$ | $O(1)$ | $O(S)$ | ✗ | ✗ |
| [28] | ✓ | ✓ | $O(1)$ | $O(S)$ | $O(S)$ | ✗ | ✗ |
| [44] | ✓ | ✓ | $O(1)$ | $O(S)$ | $O(1)$ | ✗ | ✗ |
| [40] | ✗ | ✗ | $O(m)$ | $O(1)$ | $O(S)$ | ✗ | ✓ |
| [45] | ✓ | ✓ | $O(1)$ | $O(S)$ | $O(t)$ | ✗ | ✓ |
| [46] | ✗ | ✓ | $O(m)$ | $O(S)$ | $O(t)$ | ✗ | ✓ |
| [47] | ✗ | ✓ | $O(\hat{n})$ | $O(S)$ | $O(1)$ | ✗ | ✓ |
| **Proposed** | ✓ | ✓ | $O(1)$ | $O(S)$ | $O(1)$ | ✓ | ✓ |

only proposed provides timestamp key based revocation. Our scheme is highly efficient in terms of computational cost and communication bandwidth.

### 5.1 Security analysis

We prove the security of proposed `ARIBBE` scheme under the BDH problem cryptographic assumption.

**Theorem 4.1** *Let hash functions $\tilde{H}$ and $\tilde{H}_3$ be random oracles. The `ARIBBE` scheme in Type I pairing setting is selectively secure (IND-ID-CPA) under BDH assumption if it holds in $\tilde{\mathbb{G}}$. For maximum count n of legitimate users, it renders the equation $Adv(\tau, q) \geq \frac{\epsilon}{\hat{e} \cdot n \cdot q_{enc} \cdot q_{\tilde{H}_3}}$, where $q_{enc}$ and $q_{\tilde{H}_3}$ denotes the adaptively asked queries.*

*Proof.* Let there exists an adversarial algorithm $\mathcal{A}$ that can break our proposed scheme with advantage $\epsilon$. The algorithm $\mathscr{C}$ can solve the BDH with advantage $\epsilon$ by executing algorithm $\mathcal{A}$.

Let (P, aP, bP, cP) be a random input instance of BDH selected by $\mathscr{C}$ and its aim is to evaluate $\hat{e}(P, P)^{abc}$.

1. **Setup**. Simulator algorithm computes $P_{pub}$ = aP and generates
   $PP = (\mathcal{G}, P, P_{pub}, \tilde{H}, \tilde{H}_1, \tilde{H}_2, \tilde{H}_3, \tilde{H}_4)$, where $\tilde{H}$ and $\tilde{H}_3$ are random oracles restrained by $\mathscr{C}$.

   (a). **Query I**. Challenger is suppose to respond to query for identity $ID_i$. Initially simulator has an empty tuple $\mathscr{T} = (ID_i, f_i, g_i, h_i)$. It transmits $\tilde{H}_{ID_i}$ value if $ID_i$ is in tuple, otherwise, $\mathscr{C}$ selects $g_i \in \mathbb{Z}/q\mathbb{Z}$ randomly and $f_i \in \{0, 1\}$ and $\omega = Pr[f_i = 0]$. If $f_i = 0$ then compute $h_i = bP^{g_i}$, or otherwise, evaluate $h_i = P^{g_i}$ and append the tuple to $\mathscr{T}$ and return $h_i$.

   (b). **Query II**. Simulator has an empty tuple $\mathcal{T} = (ID_i, y_i, v_i)$. It transmits $v_i = \tilde{H}_3(y_i, ID_i)$ value if $(y_i, v_i, ID_i)$ is in tuple, otherwise, $\mathscr{C}$ randomly selects $v_i \in \tilde{\mathbb{G}}$ and append the tuple $(ID_i, y_i, v_i)$ to the $\mathscr{T}$ and transmits $v_i$ to $\mathcal{A}$.

2. **Phase I**. $\mathcal{A}$ transmits decryption key queries to $ID_i$, $\mathscr{C}$ obtains the value of $f_i, g_i$ from list of tuple $\mathscr{T}$. If $f_i = 0$, $\mathscr{C}$ aborts the process, else $\mathscr{C}$ computes
   $D_{SKID_i} = s \cdot \tilde{H}_1(ID_i) = aP^{g_i} = g_i P_{pub}$.

3. **Challenge**. After completion of *Phase I*, $\mathcal{A}$ transmits two equal length distinct messages $M_0$, $M_1$ and a challenge identity set $S^* = (ID_1, ID_2, \cdots, ID_n)$ along with the only constraint that $\mathcal{A}$ cannot query the decryption key for any $ID_i \in S^*$ in Phase I. $\mathscr{C}$ randomly selects a bit $\beta \in \{0, 1\}$ and function as follows.

   (a). Select a dummy identity $ID_0 \notin S^*$ and $\tilde{\delta}_i^* \in \tilde{\mathbb{G}}$, for $i = [0, n]$.

   (b). Select $(r_1, r_2 \in_R \mathbb{Z}/q\mathbb{Z})$ and $C_1 \in \tilde{\mathbb{G}}$

   (c). Compute,

$$C_2^* = r_1^* \cdot P$$
$$C_3^* = r_2^* \cdot P$$
$$C_4^* = r_3^* \cdot P$$

(d). For $i = [0, n]$ obtain the value of $\tilde{H}(ID_i)$ and evaluate

$$
\begin{aligned}
x_i^* \quad &= \tilde{H}_1(\hat{e}(x_A)^{r_1^*} \oplus \hat{e}(x_B)^{r_1^*}, ID_i) \\
\delta_i^* \quad &= \tilde{H}_2(\hat{e}(x_A)^{r_2^*}, ID_i) \\
f_1(x) \quad &= \prod_{j=0, j \neq i}^{n} \left( \frac{x - x_j^*}{x_i^* - x_j^*} \right) \\
&= \sum_{j=0}^{n} (a_{i,j} x^j \bmod q)
\end{aligned}
$$

and compute,

$$
\begin{aligned}
Q_i^* \quad &= \sum_{j=0}^{n} a_{j,i} \delta_j^* \\
U_i^* \quad &= \sum_{j=0}^{n} a_{j,i} \tilde{\delta}_j^*
\end{aligned}
$$

$$
\mathcal{C}^* = (C_1^*, C_2^*, C_3^*, C_4^*, r_1^*, [Q_i^*]_{i=0}^{n}, [U_i^*]_{i=0}^{n})
$$

4. **Phase II**. $\mathcal{A}$ produces decryption key queries adaptively with the constraint that it cannot query the decryption key for identity $ID_i$, where $ID_i \in S^*$. $\mathcal{C}$ responds as in *Phase I*.

5. **Guess**. Eventually, $\mathcal{A}$ acknowledges with the guess $\beta \in \{0, 1\}$.

Let $\mathcal{W}_i = \tilde{H}_1(\hat{e}(x_A)^c)$. In the proposed construction, $\tilde{\delta}_i^* = \kappa + \tilde{H}_3(\mathcal{W}_i)$. The output of $\tilde{H}_3(\mathcal{W}_i)$ is not known before someone querying the value of $\mathcal{W}_i$), since, $\kappa$ is encrypted with a random value independent of $\mathcal{W}_i$). Thus, $\tilde{\delta}_i^*$ acts as a one-time pad. As per the assumption considered, the adversarial algorithm must query $\tilde{H}_3$ on $\mathcal{W}_i$). Now, $\mathcal{C}$ comes up with the solution that is in $\mathcal{T}$. Therefore, from the above analysis, we define the following probabilities:

1. The probability that $\mathcal{C}$ does not abort in private key query is
$Pr[f_i = 1, i = 1, 2, \cdots, q_{enc}] = (1 - \omega)q_{enc}$.

2. The probability that at least one of the hash value of challenge identity contains BDH is $\omega$.

3. The probability that an adversary selects identity, when $f_i = 0$ is $\frac{1}{n}$.

4. The probability of choosing the correct solution of the cryptographic assumption from $\mathcal{T}$ is greater than or equal to $\frac{1}{q\tilde{H}_3}$.

Therefore,

$$
\epsilon' \geq (1 - \omega)^{q_{enc}} \cdot \omega \cdot \frac{\epsilon}{n \cdot q\tilde{H}_3} \tag{11}
$$

this can also be written as,

$$
\epsilon' \geq \frac{\epsilon}{\hat{e} \cdot n \cdot q_{enc} \cdot q\tilde{H}_3} \tag{12}
$$

## 6 Conclusion

In this paper, a new technique called `ARIBBE` cryptosystem based on a public key framework using Type-I bi-linear map was proposed. The privacy of user's content identity is one of the primary concerns in data sharing. Hence, this paper first proposed a privacy-preserving (anonymous) revocable ID-based broadcast cryptosystem with timestamp option that facilitates broadcasters to transmit encrypted data to legitimate group participants so that revoked users will not learn anything if they all collide with each other. The proposed construction also provides an access control method in online social networks that offers one-to-one and one-to-many encrypted communication. The scheme also offers a data access control method that permits a third party to revoke any recipient identity without learning the data contents and legitimate user identities. The result indicate that the proposed scheme is extremely efficient in terms of computational cost and communication bandwidth as well as secure under CPA attack, with the ciphertext size being independent of the number of receiver identities. The proposed cryptosystem could be deployed in OSN services for distributing information to provide data access control. Security proofs show that proposed security requirements are met. However, construction of a scheme with the same parameters but without pairing is left as an open problem.

## Author Contributions

**Conceptualization:** Snehlata Yadav.

**Formal analysis:** Snehlata Yadav.

**Investigation:** Snehlata Yadav.

**Methodology:** Snehlata Yadav.

**Supervision:** Namita Tiwari.

**Validation:** Snehlata Yadav.

**Visualization:** Snehlata Yadav.

**Writing – original draft:** Snehlata Yadav.

**Writing – review & editing:** Snehlata Yadav.

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
