## [Decision Letter · Decision Letter 0]

19 Jul 2022

PONE-D-22-18072Privacy Preserving data sharing method for social media platforms.PLOS ONE

Dear Dr. Yadav,

Thank you for submitting your manuscript to PLOS ONE. After careful consideration, we feel that it has merit but does not fully meet PLOS ONE’s publication criteria as it currently stands. Therefore, we invite you to submit a revised version of the manuscript that addresses the points raised during the review process.

We look forward to receiving your revised manuscript.

Kind regards,

Pandi Vijayakumar, Ph.D

Academic Editor

PLOS ONE

Journal Requirements:

4. Please ensure that you refer to Figure 1 in your text as, if accepted, production will need this reference to link the reader to the figure.

5. Please remove your figures from within your manuscript file, leaving only the individual TIFF/EPS image files, uploaded separately.  These will be automatically included in the reviewers’ PDF.

6. Please upload a copy of Supporting Information which you refer to in your text.

Additional Editor Comments:

Based on the comments of the reviewers, I recommend this paper for major revision.

Reviewers' comments:

Reviewer's Responses to Questions

**Comments to the Author**

1. Is the manuscript technically sound, and do the data support the conclusions?

Reviewer #1: Yes

Reviewer #2: Yes

2. Has the statistical analysis been performed appropriately and rigorously? 

Reviewer #1: Yes

Reviewer #2: No

3. Have the authors made all data underlying the findings in their manuscript fully available?

Reviewer #1: Yes

Reviewer #2: Yes

4. Is the manuscript presented in an intelligible fashion and written in standard English?

Reviewer #1: Yes

Reviewer #2: Yes

5. Review Comments to the Author

Reviewer #1: Privacy Preserving data sharing method for social media platforms is discussed in this paper. This paper has not clearly shown the advantages in performance of their approach with respect to others from the literature in this field. Indeed, I found the paper a little bit difficult to read, due not only to the poor grammar used throughout, but also the unclear structure of the argument being put across. In particular, the quality of the presentation should be improved in this paper. This paper would be substantially improved by thoroughly rewriting the prose with the help of a good English-language writer. In general, this paper needs such a treatment before being considered any further. Furthermore, presentation aside, by reading the paper, it still was not entirely clear what to expect with the direction of the article. Indeed, the contribution proposed in this paper should properly be compared and contextualized with respect to state of the art. The aspects mentioned above should be carefully addressed before the paper can be considered any further. Please consider the following remarks to improve your article:

Explain the novelty of your work presented in this work.

Paper needs to polish and provide a detailed explanation of theoretical aspects such as conditions and theorems, and practical issues like algorithms, rules and possible applications.

The Introduction section needs to be re-written to improve its quality and readability.

Improve the quality of figures and explain those properly.

Following are some relevant and recent references which may be explored to improve overall quality of the revised paper:

Recurrent neural network (RNN) to analyse mental behaviour in social media,

Multiple features based approach for automatic fake news detection on social networks using deep learning,

Detecting compromised social network accounts using deep learning for behavior and text analyses,

A status property classifier of social media user's personality for customer-oriented intelligent marketing systems: intelligent-based marketing activities,

Named Entity Recognition for Code Mixed Social Media Sentences,

Research synthesis and thematic analysis of twitter through bibliometric analysis,

Secure Timestamp-Based Mutual Authentication Protocol for IoT Devices Using RFID Tags,

A novel spatio-temporal access control model for online social networks and visual verification,

Blockchain-assisted secure fine-grained searchable encryption for a cloud-based healthcare cyber-physical system,

Secure and Energy Efficient-Based E-Health Care Framework for Green Internet of Things,

IoT transaction processing through cooperative concurrency control on fog–cloud computing environment,

Many references are with incomplete bibliographic information (like lack of publication venue, for instance). This must be corrected

There are many English and grammatical issues in the paper which need to be rectified.

The formula character format is best to be different from the main text, and mathematical characters are recommended.

In the related works, "et al" should be "et al.".

It seems that the contribution points of the article are a little bit few. After or in the section of Motivation, it is recommended that the authors summarize the contribution points of their work, which clearly demonstrate the innovations.

Reviewer #2: The authors have developed a new scheme for privacy preserving for data sharing to be used in healthcare applications. The following concerns must be addressed before final publication.The comments are attached as a separate file

6. PLOS authors have the option to publish the peer review history of their article (what does this mean?). If published, this will include your full peer review and any attached files.

Reviewer #1: No

Reviewer #2: No

---

## [Author Response · Author response to Decision Letter 0]

9 Nov 2022

PLOS ONE RESPONSE SHEET

Reviewer #1: Privacy preserving data sharing method for social media platforms is discussed in this paper. This paper has not clearly shown the advantages in performance of their approach with respect to others from the literature in this field. Indeed, I found the paper a little bit difficult to read, due not only to the poor grammar used throughout, but also the unclear structure of the argument being put across. In particular, the quality of the presentation should be improved in this paper. This paper would be substantially improved by thoroughly rewriting the prose with the help of a good English-language writer. In general, this paper needs such a treatment before being considered any further. Furthermore, presentation aside, by reading the paper, it still was not entirely clear what to expect with the direction of the article. Indeed, the contribution proposed in this paper should properly be compared and contextualized with respect to state of the art. The aspects mentioned above should be carefully addressed before the paper can be considered any further.

Please consider the following remarks to improve your article:

1. Explain the novelty of your work presented in this work.

Paper needs to polish and provide a detailed explanation of theoretical aspects such as conditions and theorems, and practical issues like algorithms, rules and possible applications.

Response: We would like to thank the reviewers for their thoughtful comments and efforts towards improving our manuscript. As suggested in comment we have rewritten the motivation section to explain the novelty of the scheme. We have performed multiple proofreads and removed all the grammatical and typographical mistakes from the manuscript.

2. The Introduction section needs to be re-written to improve its quality and readability.

Response: Thank you for the comment. As per your suggestion we have rewritten the introduction section.

3. Improve the quality of figures and explain those properly.

Response: Thank you for the comment. As per your suggestion we have Improve the quality of figures and explain them.

4. Following are some relevant and recent references which may be explored to improve overall quality of the revised paper:

a. Recurrent neural network (RNN) to analyse mental behaviour in social media.

b. Multiple features based approach for automatic fake news detection on social networks using deep learning.

c. Detecting compromised social network accounts using deep learning for behavior and text analyses.

d. A status property classifier of social media user's personality for customer-oriented intelligent marketing systems: intelligent-based marketing activities.

e. Named Entity Recognition for Code Mixed Social Media Sentences.

f. Research synthesis and thematic analysis of twitter through bibliometric analysis.

g. Secure Timestamp-Based Mutual Authentication Protocol for IoT Devices Using RFID Tags.

h. A novel spatio-temporal access control model for online social networks and visual verification.

i. Blockchain-assisted secure fine-grained searchable encryption for a cloud-based healthcare cyber-physical system.

j. Secure and Energy Efficient-Based E-Health Care Framework for Green Internet of Things.

k. IoT transaction processing through cooperative concurrency control on fog–cloud computing environment

Response: Thank you for the comment. As per your suggestion we have cites these important research papers in our manuscript.

%%%%%%%%%%%%%%%%%%%%%%%%%%%%%%%%%%%%%%%%%%%%%%%%%%%%%%%%%%%%%%%%%

Reviewer #2: 

The authors have developed a new scheme for privacy preserving for data sharing to be used in healthcare applications. The following concerns must be addressed before final publication.

1. Introduction section must be strengthened exhibiting the scope and the motivations for developing the proposed system.

Response: Thank you for the comment. As suggested in comment we have revised the Introduction section to strengthen the scope of the proposed system.

2. The objectives of the proposed system are not defined clearly.

Response: Thanks for the comment. As suggested in comment we have defined the objectives of the proposed system.

3. Literature survey section must be extended. Some of the important manuscripts related to privacy preserving are not discussed. The titles of the related manuscripts are given below:

a) Computationally efficient privacy preserving anonymous mutual and batch authentication schemes for vehicular ad hoc networks. 

b) Computationally efficient privacy preserving authentication and key distribution techniques for vehicular ad hoc networks. 

c) EAAP: Efficient Anonymous Authentication With Conditional Privacy-Preserving Scheme for Vehicular Ad Hoc Networks. 

d) CPAV: Computationally Efficient Privacy Preserving Anonymous Authentication Scheme for Vehicular Ad Hoc Networks.

e) Review: Privacy-Preservation in the Context of Natural Language Processing.

f) A secure and efficient authentication and data sharing scheme for Internet of Things based on blockchain.

g) Security and Privacy-Preserving Challenges of e-Health Solutions in Cloud Computing. 

h) MGPV: A novel and efficient scheme for secure data sharing among mobile users in the public cloud. 

Response: Thank you for the comment. As suggested in comment we have extended the literature survey section and cited all these (from a-h) important papers in our manuscript.

4. The equations must be numbered that are used throughout the manuscript.

Response: Thank you for the comment. As suggested in comment we have equations are numbered. 

5. Table 2 is not visible completely.

Response: Thank you for the comment. As suggested in comment, now, Table 2 is completely visible.

6. Some graphical analysis during the comparative analysis must be shown for easy understanding.

Response: Thank you for the comment. As suggested the comparative analysis is done.

---

## [Decision Letter · Decision Letter 1]

22 Dec 2022

Privacy preserving data sharing method for social media platforms.

PONE-D-22-18072R1

Dear Dr. Yadav,

We’re pleased to inform you that your manuscript has been judged scientifically suitable for publication and will be formally accepted for publication once it meets all outstanding technical requirements.

Kind regards,

Pandi Vijayakumar, Ph.D

Academic Editor

PLOS ONE

Additional Editor Comments (optional):

Reviewers' comments:

Reviewer's Responses to Questions

**Comments to the Author**

1. If the authors have adequately addressed your comments raised in a previous round of review and you feel that this manuscript is now acceptable for publication, you may indicate that here to bypass the “Comments to the Author” section, enter your conflict of interest statement in the “Confidential to Editor” section, and submit your "Accept" recommendation.

Reviewer #1: (No Response)

2. Is the manuscript technically sound, and do the data support the conclusions?

Reviewer #1: (No Response)

3. Has the statistical analysis been performed appropriately and rigorously? 

Reviewer #1: (No Response)

4. Have the authors made all data underlying the findings in their manuscript fully available?

Reviewer #1: (No Response)

5. Is the manuscript presented in an intelligible fashion and written in standard English?

Reviewer #1: (No Response)

6. Review Comments to the Author

Reviewer #1: Privacy preserving data sharing method for social media platforms is presented in this paper. Paper is revised well and it can be considered for the publications. Moreover, a language check is recommended.

7. PLOS authors have the option to publish the peer review history of their article (what does this mean?). If published, this will include your full peer review and any attached files.

Reviewer #1: No

---

## [Editor Report · Acceptance letter]

3 Jan 2023

PONE-D-22-18072R1 

Privacy preserving data sharing method for social media platforms 

Dear Dr. Yadav:

I'm pleased to inform you that your manuscript has been deemed suitable for publication in PLOS ONE. Congratulations! Your manuscript is now with our production department. 

Kind regards, 

on behalf of

Dr. Pandi Vijayakumar 

Academic Editor

PLOS ONE